# “The Pupillary (Hippus) Nystagmus”: A Possible Clinical Hallmark to Support the Diagnosis of Vestibular Migraine

**DOI:** 10.3390/jcm12051957

**Published:** 2023-03-01

**Authors:** Mauro Gufoni, Augusto Pietro Casani

**Affiliations:** ENT Section, Medical, Molecular and Critical Area, Department of Surgical Pathology, Pisa University Hospital, 56122 Pisa, Italy

**Keywords:** pupillary hippus, pupillary nystagmus, vestibular migraine, vertigo, vestibular examination

## Abstract

(1) Background: Hippus (which in this paper will be called “Pupillary nystagmus”) is a well-known phenomenon which has never been related to any specific pathology, so much so that it can be considered physiological even in the normal subject, and is characterized by cycles of dilation and narrowing of the pupil under constant lighting conditions. The aim of this study is to verify the presence of pupillary nystagmus in a series of patients suffering from vestibular migraine. (2) Methods: 30 patients with dizziness suffering from vestibular migraine (VM), diagnosed according to the international criteria, were evaluated for the presence of pupillary nystagmus and compared with the results obtained in a group of 50 patients complaining of dizziness that was not migraine-related. (3) Results: Among the 30 VM patients, only two cases were found to be negative for pupillary nystagmus. Among the 50 non-migraineurs dizzy patients, three had pupillary nystagmus, while the remaining 47 did not. This resulted in a test sensitivity of 0.93% and a specificity of 0.94%. (4) Conclusion: we propose the consideration of the presence of pupillary nystagmus as an objective sign (present in the inter-critical phase) to be associated with the international diagnostic criteria for the diagnosis of vestibular migraine.

## 1. Introduction

Vestibular migraine (VM) is characterized by recurrent vestibular attacks that are not associated with migraine headache. VM is now considered as the first cause of episodic vertigo in adults [1], and it is a common diagnosis in children [2]. The diagnosis is primarily based on clinical history, and international guidelines have been developed [3,4]. While the presence or history of migraine is essential for its diagnosis, the headache and dizzy symptoms do not need to temporally coincide. The instrumental examination of patients with VM shows normal results or variable and inconsistent abnormalities, but vestibular testing needs to be performed with the aim of excluding other disorders considered in differential diagnosis. This implies that it is necessary to spend time collecting the detailed clinical history of the patient who, however, is not always able to describe his symptoms exactly with the risk of omitting important details for diagnostic purposes.

Having an instrumental hallmark would be extremely useful, especially in cases where the clinical picture does not fully meet the international diagnostic criteria. In this paper, a sign that has been well known for years and whose origin has never been defined with certainty, so much so that it was classically considered a phenomenon without clinical value, was taken into consideration. It is a very characteristic behavior of the pupil, which dilates and contracts cyclically in the presence of constant lighting, independently of eye movements or change in illumination [5]. It has been called “pupillary hippus” (PH), “pupillary athetosis”, or, in English-speaking countries, “pupillary unrest” or “dancing pupils”, terms that seem, however, to be rather non-specific [6]. PH usually occurs in a physiologically drowsy state, and can range from 0.04 to 2 Hz [5], and the magnitude of the pupil size variations range from not detectable to over 0.5 mm [7]. A precise definition of PH is lacking; the variety of techniques used to assess the pupil movements and the interindividual variation do not allow for validated parameters to consider PH as pathological. PH has been observed in patients suffering from epilepsy [7] or neurotic disorders [8], diabetic autonomic neuropathy [9] and is associated with disorders of the autonomic nervous system [10]. On the other hand, dysautonomia has also been reported to underlie migraine disease [11], and the presence of PH has also been reported in migraineurs [12,13]. In this paper we evaluated the presence of PH in patients suffering from VM with the aim of identifying an objective sign that may be potentially useful in helping the physician with the diagnosis of VM, especially when the criteria indicated by international guidelines are not fully met.

## 2. Materials and Methods

Two series of patients complaining of vertigo and dizziness were considered:30 patients consecutively diagnosed as suffering from definite VM (mean age 52 years, minimum 6 years, maximum 77 years, 11 males and 19 females). The diagnosis of vestibular migraine was made based on international criteria [3]. We excluded from the study patients with probable VM and subjects who had received ear or eye surgery or other significant comorbidities.50 consecutive patients (mean age 58 years, minimum 17 years, maximum 89 years, 23 males and 27 females) affected by vertigo and dizziness not attributable to VM, who constituted the control group. All the patients belonging to the control group did not suffer from migraine or other types of headaches.

The patients belonging to the control group were affected by paroxysmal positional vertigo (15), acute vestibular deficit (8), vascular vertigo (6), Meniere’s disease (6), and acoustic neuroma (1). A total of 14 patients showed a normal examination, and the dizziness was attributable to diseases that were not strictly vestibular (such as pharmacological dizziness, orthostatic hypotension, and undiagnosed PPPD).

Patients underwent a thorough medical history, otoscopy, neurological evaluation (cerebellar tests and clinical evaluation of the cranial nerves), audiometry, evaluation of the spontaneous and positional nystagmus, and a head shaking test using infrared goggles. The instrumental examination consisted of performing video-HIT, functional-video-HIT, caloric testing, and cervical and ocular VEMPs.

During our experience, we have informally begun to call PH by the heterodox expression: “Pupillary Nystagmus”‘ (PNy). It is well known that no correlation exists between the pupillomotor response and the vestibulo-ocular reflex. From a semeiological point of view, PH could have some similarity with the well-known extra-vestibular nystagmus. Nystagmus is defined as “... a repetitive to and fro movement of the eyes that includes smooth sinusoidal oscillations (pendular nystagmus)” [14]. PH could be defined inductively as “... a repetitive to and fro change in the pupil diameter that includes smooth sinusoidal oscillations”. The only difference is that this phenomenology affects intrinsic rather than extrinsic eye muscles. The semantic expression ‘pupillary nystagmus’ is intended only as a current, but suggestive, variant with the aim of referring to the common traits that the phenomenon of pupillary hippus has with extravestibular nystagmus.

The assessment of pupillary nystagmus (PNy) (presence/absence) was performed under Frenzel glasses, and a video (lasting at least 10 s) was taken. The examiner evaluated the visible amplitude of PNy during the whole observation period under Frenzel glasses. Patients entering the study did not report any other neurological or eye problems or significant head injuries. None of the patients took any drugs that could affect the autonomic nervous system. The presence or absence of pupillary nystagmus was assessed by two different examiners in a double-blind manner: each of them was unaware of the evaluations of the other, including the medical history and examination results. In no case was there a discrepancy in evaluation, demonstrating the ease of observation of the sign. All patients underwent contrast-enhanced brain nuclear magnetic resonance imaging.

A statistical evaluation was performed using a Pearson’s chi-squared test, phi coefficient and Bayesian contingency tables—BF10 [GNU Project (2015). GNU PSPP (Version 0.8.5) [Computer Software]. Free Software Foundation. Boston, MA, USA; JASP Team (2022). JASP (Version 0.16.3).

Ethical review and approval by the local Institutional Board (Comitato Etico Azienda Ospedaliero-Universitaria Pisana, Pisa, Italy) were waived for this study. Due to its retrospective nature, it was not set up as part of a research project. Furthermore, the study does not include new experimental diagnostic protocols, and the patients included in the study were diagnosed according to national guidelines. Written informed consent was obtained from all participants, and the study was conducted in accordance with the 1964 Declaration of Helsinki.

## 3. Results

Among the 30 VM patients, only two cases were found to be negative for pupillary nystagmus (Appendix A). Among the 50 non-migraineur dizzy patients, three had pupillary nystagmus, while the remaining 47 did not. Table 1 shows the results obtained in the two groups.

This resulted in a test sensitivity of 0.93% and a specificity of 0.94%.

The positive predictive value is 0.90, and the negative predictive value is 0.96.

A statistical evaluation was undertaken using the chi-square test, and showed a significant difference (X^2^ value 60.25, *p* < 0.001, Phi 0.87, BF_10_ independent multinomial 8.598 × 10^+13^). (Table 2).

## 4. Discussion

The diagnosis of VM is based quite exclusively on the history taking; there is no pathognomonic clinical sign for VM and there are no gold standard diagnostic tests for VM.

The availability of some clinical or instrumental test with a relatively high sensibility and specificity would be very useful, especially when the international criteria are not completely fulfilled. Only the functional video HIT performed with an optokinetic stimulation seems to provide some positive results in VM, indicating a visual dependence in VM patients complaining of visually induced vertigo, head motion–induced vertigo, and head motion–induced dizziness with nausea [15,16]. Usually, the diagnosis of VM is made by an audiologist, otolaryngologist or neurologist. It would be appropriate for the ideal sign associated with vestibular migraine to have all of the following characteristics:○strongly suggestive (even if not pathognomonic) of the condition, therefore present in as many VM patients as possible and absent in most patients with vertigo or dizziness not migraine related;○easily identifiable on otoneurological examination;○present in the inter-critical phase, since it is difficult to examine a VM patient in the acute stage of the disease;○ease of recording and archiving.

In the absence of changes in external influences such as luminance, mood, and fixation, the pupil is in constant motion. An exaggeration of this phenomenon is usually termed Pupillary Hippus: its frequency ranges from 0.04 to 2 Hz [5], and the magnitude of the pupil size variations usually do not surpass 0.5 mm [5]. It is more evident in pupils of medium amplitude and has a periodic pattern (the period measured was 5 seconds^5^) but the course may not be constant over time (Figure 1). The origin of pupillary hippus is believed to be related to an abnormal activity of the autonomic nervous system because it can be inhibited by pharmacologically antagonizing the parasympathetic system [5,10].

The disruption of the balance between the sympathetic and parasympathetic systems is considered a pathophysiological mechanism underlying migraine disease [13], and changes in pupillary function have been observed in migraine both in the headache attack and in the intercritical phase [17].

It has been reported that the left cerebral hemisphere is mainly involved in parasympathetic activity, and the right in sympathetic system activity. Parasympathetic stimulation in unilateral migraineurs causes significant skin phenomenology on the stimulated side, and sympathetic stimulation does not seem to influence this significantly. Activation in this case would occur through a trigemino-parasympathetic reflex, resulting in vasodilation and the increase in secretory phenomena [18]. There is evidence of a lower sympathetic activity in migraineurs, demonstrated by an increased latency to the light reflex, after apraclonidine administration [19]. Reduced nocturnal activity of the parasympathetic system has also been demonstrated in migraine patients, especially in subjects with aura [20]. Furthermore, cardiac vagal responses via baroreceptors are reduced in migraine patients, but sympathetic system-related responses are not. As a consequence, it appears that the autonomic nervous system may play a role in the pathophysiology of migraine [21]. The pupillary hippus phenomenon can be extinguished with antagonists of the parasympathetic nervous system, whereas antagonists of the sympathetic system dilate the pupils without blocking the hippus: this suggests that the phenomenon originates in the centrally localized parasympathetic system and not in the sympathetic system [10]. Furthermore, parasympathetic activity contributes to the onset of pain in migraine by activating or sensitizing (or both) the intracranial nociceptors [10].

It seems well established in the literature that [10,18,19]:the pupil reacts to asymmetries in the balance between the sympathetic and parasympathetic through changes in its diameter, with particular dependence on vagal tone;An imbalance between the sympathetic and parasympathetic can contribute to the genesis of painful migraine pathology.

It has recently been shown that the pupillary cycle has specific characteristics in the migraine sufferer: in particular, the pupillary cycle period is longer in the migraine. This data allows the differentiation of a migraine patient from a non-migraine patient [21].

The pupillary cycle is a well-known phenomenon [22], and consists in the projection of a luminous dot onto the pupil, very close to the edge of the iris. The photomotor reflex causes miosis, which prevents the light beam from reaching the retina. Consequently, the pupil dilates, and the light reaches the retina again, giving rise to a new cycle. The frequency of the pupillary cycle allows for the evaluation of the sympathetic-parasympathetic balance and, consequently, the predisposition to migraine.

Our results recorded in a group of VM patients demonstrate a very high incidence of PNy (in contrast with the low incidence observed in patients suffering from vertigo and dizziness not migraine related) whose presence could be considered as a hallmark of the disease. The high sensitivity and sensibility of PNy makes this sign highly pathognomonic of VM, and it could be helpful in patients with possible VM. We have found PNy in the only child (six years of age) present in our series of patients suffering from VM. In children, the hippus frequency seems to be higher than it is in adults, suggesting the influence of the sympathetic branch of the autonomic nervous system on it that decreases with age and maturation [23]. For this reason, the presence of PNy in children must be considered with caution. Two patients in definite VM groups did not show PNy; analyzing their clinical and instrumental characteristics, we found no difference compared from those who showed PNy.

As it is not clear in the literature whether the phenomenon is also present in the dark, it is advisable to search for the sign under Frenzel’s glasses or, in any case, in a permanently lit environment (Table 3). However, the use of a binocular system of evaluating the pupillary movements is recommended; a unilateral PH was described in migrainous patients. The search under infra-red video-Frenzel should be avoided entirely, or at least until evidence of the presence of pupillary nystagmus (even in the dark) is obtained.

The sign was always detected in the inter-critical phase (none of the patients examined were in the acute vertiginous crisis phase or reported headache at the time of observation). Little time is required for the examination, as it is to be considered exactly like the ‘bedside’ search for a spontaneous nystagmus. However, it needs to focus specifically on the pupil if one wants to avoid losing the data. In our case series, pupillary nystagmus was present in the great majority of VM patients. It was also present in three patients considered non-migraineurs (6%), but we cannot exclude that in those cases the anamnesis was lacking, given the well-known difficulty in identifying headache crisis as migraine, which is often wrongly attributed to different causes (neck pain, neuralgia, sinusitis, etc.). As an alternative to the study of the pupillary cycle which is not complicated but which requires the help of the Ophthalmologist, we propose the direct search of this sign (using Frenzel glasses) in dizzy patients, especially when a VM is suspected.

The main limitation of this study is the small size of the sample and the heterogeneity of the control group. For this reason, we are planning a study with a large number of patients evaluating additional factors such as age, gender, and course of VM, in order to allow for more significant results. Moreover, the method we have proposed for evaluating the PNy is certainly less precise than an ophthalmological study of the pupillary cycle [21]. Using an infrared pupillometer would be more accurate than assessing pupil size under Frenzel goggles. Nevertheless, as a part of the bedside assessment of patients with suspected VM, this practical and simple evaluation of the pupil movements seems to be sufficiently valuable.

## 5. Conclusions

Even if our results need to be confirmed in a larger series of patients, we propose the observation of pupillary nystagmus as an objective sign helping the physician to diagnose vestibular migraine, being very common in the intercritical phase of this pathology and rarely encountered in dizzy patients that are non-migraine sufferers. This sign is easy to observe and is recordable with a camera or smartphone. We recommend the observation of PNy whose presence could be considered as a supplementary element to reenforce the diagnosis of VM based mainly on the clinical criteria suggested by a joint committee of the International Headache Society (IHS) and the Barany Society [4].

## Figures and Tables

**Figure 1 jcm-12-01957-f001:**
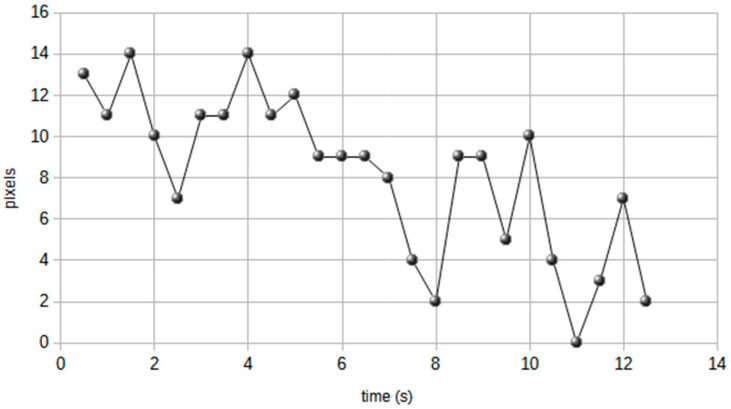
Pupil diameter trend in a migraine patient, shown by the measurement (pixels) performed on the video (by individual frame).

**Table 1 jcm-12-01957-t001:** Prevalence of pupillary nystagmus in patients with vestibular migraine and in non-migraine patients. VM: vestibular migraine; No VM: dizzy patients not suffering from vestibular migraine; PNy+ patients presenting with pupillary nystagmus; PNy− patients who do not have pupillary nystagmus.

	VM	No VM	Total
PNy+	28	3	31
PNy−	2	47	49
Total	30	50	80

**Table 2 jcm-12-01957-t002:** Bayesian Contingency Tables. Pny: pupillary nystagmus.

Contingency Tables			
	Migraine		
Pny	0	1	Total
0	47	2	49
1	3	28	31
Total	50	30	80
Bayesian Contingency Tables Tests			
			**Value**
BF_10_ Independent multinomial			8.496 × 10^+13^
N			80

Note. For all tests, the alternative hypothesis specifies that group 0 is not equal to 1.

**Table 3 jcm-12-01957-t003:** The modalities of observation of PNy.

The patient is examined with the eyes open under Frenzel glasses
2. Lighting must be constant
3. Vergence movements and blink reflex should be avoided
4. The observation must last at least 10 s
5. A video recording of PNy is recommended

## Data Availability

Data are contained in the article.

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
