# Peer review of "“The Pupillary (Hippus) Nystagmus”: A Possible Clinical Hallmark to Support the Diagnosis of Vestibular Migraine"

_jcm, 2023, doi:10.3390/jcm12051957_

Round 1
Reviewer 1 Report
Studying the pupillary hippus as an objective for vestibular migraine and migraine seems to be an interesting approach. However, I have some major concerns regarding the study design.
- Presence or absence of migraine in the control group was not systematically assessed. I would expect pupillary hippus to be a hallmark of migraine in general and not vestibular migraine in particular.
- A definition of pupillary hippus is missing (i.e. when is a variation in pupil size considered pupillary hippus and when is it not).
. Why have the authours chosen the term "pupillary nystagmus" rather than pupillary hippus, which is commonly used?
- Using an infrared pupillometer would be more accurate than assessing pupil size under Frenzel goggles. The hyperlink to view the supplementary material is not working.
- A prospective study design would be fairly easy to achieve given the common diagnosis of vestibular migraine and large number of dizzy patients
Author Response
Studying the pupillary hippus as an objective for vestibular migraine and migraine seems to be an interesting approach. However, I have some major concerns regarding the study design.
We thank the reviewer for its valuable comment and for the interest in this new method of evaluation of patients suffering from vestibular migraine. We tried to improve the study design
- Presence or absence of migraine in the control group was not systematically assessed. I would expect pupillary hippus to be a hallmark of migraine in general and not vestibular migraine in particular.
We agree with the reviewer’s comment. Obviously, all the patients of the control group were investigated about the presence of migraine and any subject suffered from migraine headache. We apologize for omitting this important clinical data and for this reason we have added a sentence in the text for clarity
- A definition of pupillary hippus is missing (i.e. when is a variation in pupil size considered pupillary hippus and when is it not).
We agree with this statement. A precise definition of PH is lacking; the variety of techniques used to assess the pupil movements and the interindividual variation do not allow to have validated parameters to consider PH as pathological. For this reason, we added a sentence both in the introduction and in the study limitations section of the manuscript.
. Why have the authours chosen the term "pupillary nystagmus" rather than pupillary hippus, which is commonly used?
We propose to introduce the term “pupillary nystagmus” instead than PH with the aim to underline its similarity to nystagmus commonly detectable in dizzy patients: we could consider PH as an objective phenomenon consequent form an activation of the intrinsic ocular muscles as well as nystagmus is provoked by the activation of the extrinsic ocular muscles by the vestibulo-ocular reflex. We added this sentence in the result section
- Using an infrared pupillometer would be more accurate than assessing pupil size under Frenzel goggles. The hyperlink to view the supplementary material is not working.
- A prospective study design would be fairly easy to achieve given the common diagnosis of vestibular migraine and large number of dizzy patients
We agree with these observations, and we have underlined these considerations in the study limitation section of the manuscript. WE also corrected the iperlink the the supplementary material
Reviewer 2 Report
The topic of this paper is extremely interesting, and it introduces a new possible sign to check for during a bedside examination. Moreover, if hippus would be confirmed as pathognomonic for VM in larger cohort studies, it may help clinicians identify VM patients, especially when the subjects do not meet all of the criteria for the diagnosis.
- Did all of the patients fulfilled the criteria for VM or were there patients with “probable” VM?
- “30 patients consecutively diagnosed as suffering from VM (mean age 52 years, 57 minimum 6, maximum 77, 11 males and 19 females)”
- One patient was very young (6-year-old). Did he present hippus?
There were any other paediatric patients?
- Hippus has also been described in the literature as a sign of multiple sclerosis and epilepsy. Was imaging and/or EEG performed? If so, what were the outcomes?
- What were the characteristics of the two patients without hippus in the VM cohort?
- Did PNy patients have any additional comorbidities?
Further research with a large cohort of patients, evaluating additional factors such as age, gender, and course of VM, is required. If the hypothesis is validated, a revision of VM diagnostic criteria is desirable.
Author Response
The topic of this paper is extremely interesting, and it introduces a new possible sign to check for during a bedside examination. Moreover, if hippus would be confirmed as pathognomonic for VM in larger cohort studies, it may help clinicians identify VM patients, especially when the subjects do not meet all of the criteria for the diagnosis.
We thank the reviewer for its valuable comment and for the interest in this new method of evaluation of patients suffering from vestibular migraine
- Did all of the patients fulfilled the criteria for VM or were there patients with “probable” VM?
We apologize for omitting this important clinical data: all the patients in the study group suffered from definite vestibular migraine. We pointed out this data in the materials and method section of the manuscript.
- “30 patients consecutively diagnosed as suffering from VM (mean age 52 years, 57 minimum 6, maximum 77, 11 males and 19 females)”
- One patient was very young (6-year-old). Did he present hippus?
There were any other paediatric patients?
Yes, this child showed pupillary nystagmus and in our series this patient was the only child. We added a sentence to clarify the composition of our group of patients suffering from vestibular migraine and we discus about the difference of pupillary hippus between children and adult
- Hippus has also been described in the literature as a sign of multiple sclerosis and epilepsy. Was imaging and/or EEG performed? If so, what were the outcomes?
All the patients in the two groups did not suffer from epilepsy and for this reason they did not perform EEG; furthermore, they underwent to Brain MRI excluding multiple sclerosis.
- What were the characteristics of the two patients without hippus in the VM cohort?
Two patients in definite VM groups did not show PNy: analyzing their clinical and instrumental characteristics, we found no difference compared from those who showed PNy.
- Did PNy patients have any additional comorbidities?
No other significant comorbidities were evident in PNy patients
Further research with a large cohort of patients, evaluating additional factors such as age, gender, and course of VM, is required. If the hypothesis is validated, a revision of VM diagnostic criteria is desirable.
We agree with this comment
Round 2
Reviewer 1 Report
I would like to respond to some of the comments and raise two additional comments at the end.
A definition of pupillary hippus is missing (i.e. when is a variation in pupil size considered pupillary hippus and when is it not).
We agree with this statement. A precise definition of PH is lacking; the variety of techniques used to assess the pupil movements and the interindividual variation do not allow to have validated parameters to consider PH as pathological. For this reason, we added a sentence both in the introduction and in the study limitations section of the manuscript.
I understand there is no commonly accepted definition. However, authors need to state based on which criteria they defined whether hippus was present or not (visible amplitude? intermittently present or during the whole observation period?)
Why have the authours chosen the term "pupillary nystagmus" rather than pupillary hippus, which is commonly used?
We propose to introduce the term “pupillary nystagmus” instead than PH with the aim to underline its similarity to nystagmus commonly detectable in dizzy patients: we could consider PH as an objective phenomenon consequent form an activation of the intrinsic ocular muscles as well as nystagmus is provoked by the activation of the extrinsic ocular muscles by the vestibulo-ocular reflex. We added this sentence in the result section.
This explanation seems is misleading. I do not know of any evidence the pupillomotor response is influenced by the vestibulo-ocular reflex. Please can the authors re-phrase or choose a term different from pupillary nystagmus.
Line 203: was there one child or several children?
Lines 227-233: Please check semantics of newly added sentences
Author Response
I would like to respond to some of the comments and raise two additional comments at the end.
A definition of pupillary hippus is missing (i.e. when is a variation in pupil size considered pupillary hippus and when is it not).
We agree with this statement. A precise definition of PH is lacking; the variety of techniques used to assess the pupil movements and the interindividual variation do not allow to have validated parameters to consider PH as pathological. For this reason, we added a sentence both in the introduction and in the study limitations section of the manuscript.
I understand there is no commonly accepted definition. However, authors need to state based on which criteria they defined whether hippus was present or not (visible amplitude? intermittently present or during the whole observation period?)
We thank the reviewer for this right observation. We have added a sentence in the methods section in order to clarify the parameters of observations of pupillary hippus
Why have the authours chosen the term "pupillary nystagmus" rather than pupillary hippus, which is commonly used?
We propose to introduce the term “pupillary nystagmus” instead than PH with the aim to underline its similarity to nystagmus commonly detectable in dizzy patients: we could consider PH as an objective phenomenon consequent form an activation of the intrinsic ocular muscles as well as nystagmus is provoked by the activation of the extrinsic ocular muscles by the vestibulo-ocular reflex. We added this sentence in the result section.
This explanation seems is misleading. I do not know of any evidence the pupillomotor response is influenced by the vestibulo-ocular reflex. Please can the authors re-phrase or choose a term different from pupillary nystagmus.
We agree with this criticism. The sentence we have actually inserted could imply that there is a relationship between pupillary motility and the vestibulo-ocular reflex, which is absolutely not true.
We have informally begun to call this sign by the heterodox expression: 'pupillary nystagmus'. Undoubtedly the pupillary hippus semeiologically has much in common with the well-known extra-vestibular nystagmus. If nystagmus is defined as " ... a repetitive to and fro movement of the eyes that includes smooth sinusoidal oscillations (pendular nystagmus)1" this sign could be defined inductively as " ... a repetitive to and fro change in the pupil diameter that includes smooth sinusoidal oscillations (pendular nystagmus)". The only difference is that this phenomenology affects intrinsic rather than extrinsic eye muscles. It is understood that the semantic expression 'pupillary nystagmus' is intended only as a current, but suggestive, variant to refer to the common traits that the phenomenon of pupillary hippus has with extravestibular nystagmus. On the other hand, Anglo-Saxon semantics does not seem to offer more expressive lemmas ('Pupillary unrest' or 'Dancing pupils')
